# The Role of Graphene Oxide in the Exothermic Mechanism of Al/CuO Nanocomposites

**DOI:** 10.3390/molecules27217614

**Published:** 2022-11-06

**Authors:** Jiaxin Su, Yan Hu, Bin Zhou, Yinghua Ye, Ruiqi Shen

**Affiliations:** 1School of Chemistry and Chemical Engineering, Nanjing University of Science and Technology, Nanjing 210094, China; 2Micro-Nano Energetic Devices Key Laboratory, Ministry of Industry and Information Technology, Nanjing 210094, China

**Keywords:** graphene oxide, metastable intermixed composites (MICs), nanothermite, thermokinetics, combustion

## Abstract

Metastable intermixed composites (MICs) have received increasing attention in the field of energy materials in recent years due to their high energy and good combustion performance. The exploration of ways of improving their potential release of heat is still underway. In this study, Al–CuO/graphene oxide (GO) nanocomposites were prepared using a combination of the self-assembly and in-suit synthesis methods. The formulation and experimental conditions were also optimized to maximize the exothermic heat. The DSC analysis shows that the addition of the GO made a significant contribution to the exothermic effect of the nanothermite. Compared with the Al–CuO nanothermite, the exothermic heat of the Al–CuO/GO nanocomposites increase by 306.9–1166.3 J/g and the peak temperatures dropped by 7.9–26.4 °C with different GO content. The reaction mechanism of the nanocomposite was investigated using a DSC and thermal reaction kinetics analysis. It was found that, compared with typical thermite reactions, the addition of the GO changed the reaction pathway of the nanothermite. The reaction products included CuAlO_2_. Moreover, the combustion properties of nanocomposite were investigated. This work reveals the unique mechanism of GO in thermite reactions, which may promote the application of carbon materials in nanothermite.

## 1. Introduction

For decades, metastable intermixed composites (MICs) have attracted more and more interest in the area of energetic materials (EMs) [1,2,3,4,5]. EMs are critical to the advancement of microscale energy-demanding systems such as propulsion units, actuation parts, power and igniters [6,7,8,9]. MICs are widely used in propellants, the combustion synthesis of sophisticated materials, and high explosives due to their extraordinary energy density, great ignition performance, and high burning rate [10,11,12,13,14]. Generally, MICs consist of a metal fuel (aluminum (Al), boron, magnesium, etc.) and an oxidizer (copper oxide (CuO), bismuth trioxide, ferric oxide, etc.), and at least one of them is on the nanometer scale [9,15,16,17,18]. Among them, Al/CuO has received much attention due to its high energy release [19]. The properties of Al/CuO are actually influenced by its particle morphology [20,21]. Chen et al. [22] prepared three different morphologies of Al/CuO nano particles (NPs). By changing the nanoparticle morphology, the properties of the Al/CuO nanothermite changed significantly. Therefore, the performance of nanothermite agents can be improved by controlling the particle morphology.

Nanoscale particles are highly reactive materials and are desirable for better combustion efficiency. However, their small particle size also poses some challenges, such as their tendency to agglomerate, which leads to a reduction in specific surface area [23] and consequently affects the heat release of the nanothermite. In order to solve above problem, the immobilization of nanoparticles on certain substrates has been widely used to avoid agglomeration [24]. Graphene oxide (GO) nanoflakes have a large surface area and abundant oxygen-containing groups, which are increasingly chosen as suitable substrates for anchoring NPs to prevent their aggregation [25,26,27,28]. GO is usually prepared by chemical oxidation and the flaking of graphite powders, which has been widely researched in the area of energetic materials, biology, and chemistry [29,30]. It has surface activity and can reduce interfacial energy because its basal plane has abundant oxygen-containing groups (hydroxyl, carboxylic, epoxide, etc.) [31,32,33]. The existence of these functional groups makes it possible for GO to be functionalized with other materials [34,35,36]. In addition, GO is thermally unstable, which means it can release a significant amount of heat with only a little heat excitation. When a heavy GO chunk is heated on a heating source, it has the potential to explode in a matter of seconds [28,37]. Rajagopalan et al. [38] prepared Al-Bi_2_O_3_/graphene sheet composites using a self-assembly method, and the energy release of the nanothermite was enhanced from 739 to 1421 J/g. However, the large particle size of the oxide particles and the high loss during the self-assembly process led to the incomplete release of potential heat from the MICs.

In this work, Al–CuO/GO nanocomposites were prepared using a combination of in-suit synthesis and self-assembly routes using Cu(CH_3_COO)_2_·H_2_O as the copper source, and the experimental conditions and formulations were optimized for investigation. The nanocomposites and their reaction products were characterized by various means, such as X-ray diffraction (XRD), energy dispersive spectrum (EDS), transmission electron microscopy (TEM), etc. Moreover, the thermal behavior of the nanocomposites was studied using a differential scanning calorimeter (DSC), and their reaction kinetics were investigated. Finally, the combustion properties of the nanocomposite were studied.

## 2. Results and Discussion

### 2.1. Formula Optimization

#### 2.1.1. Effect of Al/CuO Equivalence Ratios

By adjusting the quality of the added raw materials, a series of Al–CuO/GO nanocomposites with different Al:CuO equivalent ratios were prepared. The GO content of the nanocomposites was 5 wt%. The XRD results are shown in Figure 1. In Figure 1a, the characteristic diffraction peaks of the GO (001) crystal plane appear at about 2θ = 10°. The characteristic diffraction peaks of the CuO (−111) and (111) crystal planes corresponding to the monoclinic system appeared at 35.2° and 38.5°, respectively. In addition, the corresponding characteristic diffraction peaks of the Al (111), (200), (200), and (311) crystal planes appeared at 38.5°, 44.7°, 65.1°, and 78.2°, respectively. Nevertheless, the disappearance of the reflection peak (001) of GO in the composite may demonstrate that the ruled lamellar pattern of the GO had been disrupted, and that exfoliated GO sheets had formed due to the loading of Al particles and the growth of CuO nanocrystals. No significant Al peaks were detected in the nanocomposites due to the small amount of Al in the system, as shown in Figure 1b. As the Al content grew, the characteristic Al peaks could be identified.

Furthermore, we observed of the morphology and dimensions of the nanostructures in the composites using SEM. Figure 2 shows the micrographs of the GO, different proportions of Al to CuO nanocomposites, including a partially enlarged image. As displayed in Figure 2a, the initial GO sheets were large and smooth with a thickness of about 2.2 nm. After the reaction, the whole GO was divided into a small piece of GO, and the layer became rough. As the Al content rose, the agglomeration of Al particles on the GO layer increased. The dispersion of the loaded particles on the GO was not affected by the change in CuO and Al equivalence ratios. A large amount of Al can cause particle agglomeration. Figure 2f is the local enlarged image of the composite of CuO:Al = 1.5:1. Here it can be seen more clearly that the complete GO layer was divided into small pieces with a thickness of about 0.4 nm. The SEM results show that Al and CuO nanoparticles were successfully loaded onto the surface of the GO lamellae. Figure 3 shows energy-dispersive X-ray spectroscopy (EDS) images of the nanocomposite (CuO:Al = 1.5:1). The EDS images indicate that the Al and CuO were uniformly spread in the composite particles.

In order to further observe the morphology of the composites, they were characterized by transmission electron microscopy (TEM). The composites without GO were prepared by this method as the control (Figure 4b). The Al particles were observed to be regular spheres with clear edges and a particle size of about 70–100 nm. The CuO NPs were clastic, with a particle size of 5–10 nm, and they were wrapped on the surface of the Al particles. Figure 4c is a TEM image of the Al–CuO/GO with a CuO:Al ratio of 1.5:1. The result shows that after adding GO to the reaction, compared with the smooth surface of the initial GO with a width of about 5 μm in Figure 4a, the GO layer was covered with particles, and the particles loaded on the GO layer were uniform. The lamellae were destroyed and divided into smaller lamellae with a width of about 1.5 μm. From the partially enlarged image (Figure 4d), it can be seen more clearly that the CuO particles and the Al particles were distributed uniformly over the GO sheets, and that the CuO was wrapped with Al particles, showing the same shape as the composites without GO (Figure 4b). It is notable that, except for the CuO particles on the surface of the GO sheets, some of the CuO particles were inserted into the GO sheets. This is because Cu^2+^ ions are not only adsorbed on the surface of GO, but also intercalated GO sheets. The CuO crystals then nucleate and grow, which causes the GO lamella to peel off.

The results of the SEM and TEM show that the Al and CuO particles loaded uniformly in the composites. This indicates that it is feasible to prepare Al–CuO/GO MICs using this method in the water–isopropanol system. Changing the equivalence ratio of Al–CuO had no noticeable effect on the particle loading. As the aluminum particle content increased, agglomeration appeared, which indicates that too much Al content is detrimental. However, too little Al will affect the heat release of the system. Therefore, the optimal ratio needs to be further analyzed using a DSC.

Figure 5 presents the DSC results of Al–CuO/GO MICs with different equivalence ratios of Al and CuO, and the parameters are shown in Table 1. The results show that there was one heat-absorbing peak and three exothermic peaks at 25–1000 °C. As the Al content increased, the heat release first increased and then decreased. When CuO:Al = 1.5:1, the maximum heat release was obtained. The reason is that too much Al content causes the particles to agglomerate, and the reaction does not complete. Too little Al can prevent the CuO reaction from completing. This is also consistent with the SEM results (Figure 2) analyzed previously.

#### 2.1.2. Effect of the GO Content

In this section, all composites have an equivalent ratio of CuO to Al of 1.5:1. The samples were named “Al–CuO/GO_0.5wt%_”, “Al–CuO/GO_1wt%_”, “Al–CuO/GO_3wt%_”, and “Al–CuO/GO_5wt%_”, according to the mass fractions of GO. Figure 6 shows the SEM images of the nanocomposites with varying GO content. Unlike the equivalence ratio of Al/CuO, changing the GO content can affect the loading of the particles. The pure Al–CuO (Figure 5a) are spherical, which is consistent with the TEM images (Figure 4b). It is obvious that with the increase in GO, the load of the particles on the GO layer decreased. This is because in the water–isopropanol system, GO is dispersed, the same content of particles are dispersed on more GO, and the corresponding particles on the GO layer will be reduced. This reduces the contact area between Al and CuO. The results show that the content of GO had an important effect on the distribution of Al and CuO particles on the lamella. The effect of GO content on the heat release of the composites needs to be further analyzed using a DSC.

Figure 7 shows the DSC curves of the composites with different GO contents, and the parameters are shown in Table 2. This can be observed in the range 25–1000 °C, with one heat-absorbing peak and two exothermic peaks. The exothermic peak at around 190 °C disappeared, unlike the peak for GO content of 5% in Figure 5. For thermite (GO: 0wt%), there were two exothermic peaks at 576.3 °C and 727.2 °C. Aluminum melts at 660 °C. There was a solid–solid reaction peak between Al NPs and CuO NPs before the melting point of aluminum. There was a solid–liquid reaction peak after the melting point of aluminum. Specifically, the solid–solid reaction peak occured between 550~580 °C, while the solid–liquid reaction peak occured between 700~900 °C. It can be seen from the Table 2 that the lower the GO content, the greater the heat release of the composite. When the GO content was 0.5 wt%, the heat release reached 1567 J/g. This was 1166 J/g higher than that of pure thermite (GO: 0wt%). There was a 128 J/g increase in heat release compared with the nanocomposites prepared using the single self-assembly method. According to the SEM analysis results (Figure 5), the lower the GO content, the more particles loaded onto it. This is why the heat release increased with the decrease in GO content. Moreover, with the addition of GO, the exothermic peak of thermite reaction was advanced to varying degrees, and the maximum advance was about 27 °C.

The DSC curves indicate that the exothermic and exothermic efficiency of the composite was much higher than that of pure thermite. For the purpose of understanding the reaction mechanisms of other reactions of Al–CuO/GO MICs, a series of control experiments were carried out.

CuO–GO composites were prepared using the same method. The raw GO and CuO–GO composites were tested using a DSC (Figure 8). The results show that the peak temperatures of the exothermic peaks of the GO and CuO–GO were 217.3 °C and 270 °C, respectively. This corresponds to peak 1 and peak 2 of Figure 5. It can be observed from the DSC parameters that the peak temperature of the Al–CuO/GO MICs was advanced to a certain extent. There is no GO exothermic peak at about 190 °C in Figure 7. This may be due to the fact that not all of the GO participates in the reaction of forming composites when the GO content is too high. Figure 9 shows the XRD pattern of the CuO–GO reaction product. The diffraction peaks of the graphite, CuO, Cu_2_O, and Cu can be observed. It can be inferred that some of the CuO reacted with the GO, while the other part did not participate in the reaction.

According to the previous analysis, we can conclude that the thermal reaction of the nanocomposites at about 570 °C was not a single thermite reaction. Figure 10 shows the reaction products of the nanocomposites (CuO:Al = 1.5:1, GO: 0.5 wt%) as characterized by XRD. Figure 10 shows the peaks of the CuAlO_2_, Cu_2_O, Cu, α-Al_2_O_3_, and γ-Al_2_O_3_. In addition, most alumina is likely to be amorphous, with wide peaks such as the one displayed in Figure 10. Depending on the products observed, the entire reaction can be described by Equation (1). In addition, CuAlO_2_ can be created by the reactions of Cu_2_O with Al_2_O_3_ (as illustrated in Equation (2)) [39,40].
(1)$$5\mathrm{Al}+9\mathrm{CuO}\to {6\mathrm{Cu}+\mathrm{Cu}}_{2}O+2{\mathrm{Al}}_{2}O_{3}+{\mathrm{CuAlO}}_{2}$$
(2)$${\mathrm{Cu}}_{2}O+{\mathrm{Al}}_{2}O_{3}\to 2{\mathrm{CuAlO}}_{2}$$

### 2.2. Optimization of Experimental Conditions

There are many factors affecting the crystalline growth of CuO nanoparticles, among which the amount of added water and the holding time are the most important conditions affecting the crystal growth. In this experiment, the composites were prepared with different holding times (1 h) and deionized water content (10 mL), which were single variables. The crystalline growth state and morphology of the nanoparticles were investigated and compared with the nanocomposites made under the previous conditions.

The XRD patterns of the nanocomposites with a holding time of 1 h and 10 mL of deionized water are presented separately in Figure 11a,b. In Figure 11a, the XRD patterns demonstrate intense diffraction peaks at 38.5°, 44.7°, 65.1°, and 78.2°, indicating Al in the nanocomposite. In addition, the diffraction peaks at 35.4° and 38.5° indicate CuO in the nanocomposite. Compared with Figure 1, for which the holding time was 30 min, there was no significant difference in the XRD patterns. By contrast, the XRD patterns in Figure 11b demonstrate intense diffraction peaks at 36.4°, 42.3°, 61.3°, and 73.5°, indicating Cu_2_O in this nanocomposite. This may be due to the chemical reaction between the Al nanoparticles and the excessive water forming Al(OH)_3_ and H_2_, which reduces CuO to Cu_2_O under heating conditions. Aluminum hydroxide is unstable under heating conditions and further decomposes into alumina. This process is represented by Equations (3)–(5). These results suggest that the amount of water added has a significant influence on the formation of the composites.
2Al + 6H_2_O→2Al(OH)_3_ + 3H_2_↑(3)
2CuO + H_2_→Cu_2_O + H_2_O(4)
2Al(OH)_3_→Al_2_O_3_ + H_2_O(5)

From the SEM analysis (Figure 12a), it can be seen that the CuO particles loaded on the GO lamella are obviously larger than those in Figure 2f, and that the distribution uniformity is poor. The reason for this phenomenon may be that CuO particles were loaded onto the active sites of the GO lamellae, forming nuclei and crystallizing growth in the reaction. With the increase in holding time, the CuO particles grew up gradually, resulting in an uneven distribution of CuO particles on the GO surface. In addition, as can be seen in Figure 12b, the GO lamellae were also thin, but they were loaded with bulk crystals that agglomerated on the GO lamellae. From the XRD characterization of Figure 11b, it can be inferred that these bulk crystals may be Cu_2_O crystals.

The crystallinity, grain size, and pattern of the sample were further inspected using TEM. The TEM images of the nanocomposite with a holding time of 1 h and 10 mL of deionized water are displayed in Figure 13. It can be observed clearly that the particles are no longer regular circles (Figure 13b), which is due to the reaction between part of the Al and the excessive water. Moreover, the particle size of the nanocomposites in Figure 7a can be estimated to be on the microscale, with particle sizes of approximately 0.6 μm to 1.5 μm. This means that an increase in holding time will increase the diameter of the inert particles.

Figure 14 shows the DSC curves obtained at a heat-up rate of 10 °C/min in argon, and the corresponding parameters are listed in Table 3. Compared with Figure 7, the composites with a holding time of 1 h (Figure 14a) showed no obvious difference in peak temperature. However, the heat decreased with the increased holding time. As the holding time increased from 30 min to 1 h, the heat changed from 1045.03 J/g to 652.34 J/g. This means that the increase in crystal size was not conducive to the reaction at this stage. Furthermore, the composites with a dosage of deionized water of 10 mL had no peak 2 and peak 3, meaning that there was no thermite reaction. This is the same as the findings for the XRD patterns (Figure 11b).

### 2.3. Thermokinetics Study

Compared with other multiple heating rate methods, the American Standard Testing Society (ASTM) E-689 method [37] is more commonly applied to identify the activation energy, as in Equaiton (6):(6)$$\ln\left(\beta /T_{p}^{2}\right)=\ln\left(\mathrm{AR}/E\right)-E/{\mathrm{RT}}_{P}$$
where β is the rate of heating, T_P_ is the peak temperature obtained from the DSC parameters, A is the pre-exponential factor, E is the activation energy, and R is the gas constant (8.314 J/mol^.^ K).

The ASTM E-689 method involves a free kinetics model, where the model necessitates DSC data for at least three experiments at various heating rates. By constructing and computing a least square “best fit” line using these points, the plot lgβ against 1/T_p_ was be obtained, where T_p_ was the calibrated peak temperature in K. According to the designation of E698-11, E and A are accessible from Equations (7) and (8), separately. The slope of the “best fit” line was taken as the value of dlg(β)/d(1/T_p_).
(7)$$E=-2.19R\left[\frac{\mathrm{dlg}\left(\beta \right)}{d\left(1/T_{p}\right)}\right]\;$$
(8)$$A=\frac{\beta E}{{\mathrm{RT}}_{p}^{2}}\exp\left(\frac{E}{{\mathrm{RT}}_{p}}\right)$$

The thermal behavior of the Al–CuO/GO under various heating rates is depicted in the DSC curves shown in Figure 15. As the heating rate grew, the DSC curves of the Al–CuO/GO gradually moved towards higher temperatures. As is shown in Table 4, with increasing heating rates, all characteristic temperatures will be raised, including the start temperature (T_e_), the peak temperature (T_p_), and the final temperature (T_f_).

The process of applying the ASTM E-689 method to the DSC data by the use of software is depicted in Figure 16. The activation energy of the Al–CuO/GO obtained from the slope of the plot was 150 ± 17 KJ/mol and the value of ln(A) was also calculated as 6.85 min^−1^.

The thermal reactions of energetic materials are interactions that occur in solid heterogeneous systems. The reaction progress for energy-containing composites is not normally obtained in one step, but in multiple steps. There is a complex relationship between each step. Each step may have a distinct dynamic model function. Hence, the conventional methods are unable to accurately characterize the kinetics of this complicated system. Non-linear multiple regression is an important method for deriving kinetic models. It is a unique way to determine between various reaction models and to obtain a global model that provides credible results over the entire range of parameters.

As indicated in Figure 17, the simulation of the kinetic model using NETZSCH Thermokinetics was well suited to the experimental data. The optimized values of the kinetic parameters are presented in Table 5. The value of the correlation coefficient was 0.974. Based on the calculated outcomes, the kinetic model for Al–CuO/GO was f(α) = (1 − α) ^n^(1 + k_cat_·α), where k_cat_ is the autocatalytic kinetic rate constant. The reaction order (n) of Al–CuO/GO was 1.20, and lgk_cat_ was 1.67.

### 2.4. Combustion Performance

Figure 18 shows the photographs of the combustion process of the Al–CuO/GO with different GO contents. All samples were successfully ignited, and self-sustained combustion occurred after laser irradiation. The burning rate of the samples was defined as the average combustion rate from the beginning to the end of combustion (r = ∆x/∆t). Each sample was measured at least three times, and the average value was taken as the burning rate of the sample to ensure the reliability of the data. The results are shown in Figure 19. It can be seen that the Al/CuO sample burned most vigorously and had the highest burning rate. With increasing GO content, the burning rate of the nanocomposite tended to decrease. The burning rate of the composite at 5wt% GO content was 80.6 m/s, which was 35.5% lower than that of pure Al–CuO. This indicates that GO has a negative effect on the burning rate of thermite. This result is very interesting. Because the high thermal conductivity of carbon materials can improve the thermal conductivity of composites, this may be an important factor affecting the combustion properties of the material. However, the obtained experimental results showed the opposite. This may be due to the poor thermal conductivity of rGO, a combustion product of GO. During the combustion process, rGO is deposited on the combustion end face, which hinders the subsequent convective heat transfer process.

## 3. Experimental

### 3.1. Materials and Reagents

Nano aluminum (nano-Al, 50 nm, 99.9%, Aladdin, Shanghai, China) and GO (Aladdin, Shanghai, China) were purchased for this research. The active aluminum content was 65.7%. The Isopropanol (C_3_H_8_O), ethanol (C_2_H_6_O), and Cupric Acetate Monohydrate (Cu(CH_3_COO)_2_·H_2_O) were of analytical purity, so there was no need to reprocess them, and they were bought from Sinopharm Chemical Reagent (Shanghai, China). Sodium dodecyl sulfate (SDS, Ling Feng Chemical Reagent, Shanghai, China) served as the dispersant. Distilled water was utilized during the entire working process.

### 3.2. Preparation of the Al–CuO/GO

In order to obtain the GO-modified Al–CuO with the best performance, we investigated a series of conditions and their effects on the Al–CuO/GO nanocomposites. These included the equivalent ratio of CuO:Al (2.5:1, 2:1, 1.5:1, and 1:1) and the content of GO (0 wt%, 0.5 wt%, 1 wt%, 3 wt%, and 5 wt%), in which the ratio of CuO and Al refers to the molar ratio. At the same time, since the morphology of the CuO particles was closely related to the experimental conditions, the effects of the holding time and the amount of added deionized water were studied. These were single variables during the experiment. Typically, the copper acetate monohydrate (Cu(OAc)_2_•H_2_O), Al, and GO were dissolved in 10 mL of isopropanol by sonicating for 30 min. Then, the Cu(OAc)_2_•H_2_O and Al dispersions were dispersed ultrasonically for 20 min. After this, the GO dispersion was placed in a round-bottom flask (part of the reflux equipment) and the mixed dispersion of Cu(OAc)_2_•H_2_O and Al was dropped into the GO dispersion under stirring. The mixture was heated to about 80°C under vigorous stirring and held at that temperature for 40 min. Then, 5 mL of deionized water was quickly poured into the above boiling solvent and heated at 80 °C for another 30 min. The solution was cooled to room temperature and scoured with ethanol. After this, it was dried overnight under vacuum at 55 °C.

### 3.3. Characterization of the Al–CuO/GO

The crystal structure was ascertained by X-ray diffraction (XRD) employing a Bruker D8 Advance X-ray diffractometer with Cu-Kα radiation. The grazing angle was kept at 1° and the 2θ collection angle ranged between 5° and 85° in 0.02° steps with length of stay of 1 s per point. The samples were measured by transmission electron microscopy (TEM) with a JEOL JEM-2100 microscope at 200 kV. One drop of the sample dispersion was deposited onto a 300 mesh copper grid coated with a carbon layer. The SEM images were acquired using a Quanta 250 field emission scanning electron microscope (FEI, America) outfitted with an energy-dispersive X-ray spectroscopy (EDX) detector.

Differential scanning calorimetry (DSC) analyses were performed on a TA-DSC-Q20 in the range 25–1000 °C at a heating rate of 10 °C/min under a flow of argon. Based on the DSC date, the thermokinetic properties of material were studied using NETZSCH Thermokinetics 3. The kinetic parameters were calculated using the Kissinger Method. It was feasible to determine the kinetic parameters, such as the pre-exponential factor (A), activation energy (Ea), and kinetic model (f(α)), to completely describe the entire reaction.

### 3.4. Burning Rate Measurements

In order to study the effect of GO on the combustion performance of Al–CuO, the sample was pressed into a 2 mm diameter quartz glass tube and ignited with a pulsed laser (Nd: YAG, 60 mJ, 6.5 ns). The combustion process was recorded by high-speed photography at 500,000 frames per second (fps). To ensure the reliability of the data, each sample underwent the experimental process three times.

## 4. Conclusions

In summary, Al–CuO/GO nanocomposites were prepared successfully using a combination of the self-assembly and in-suit synthesis methods. When the equivalence ratio of CuO to Al was 1.5:1 and the mass fraction of GO was 0.5%, the Al and CuO nanoparticles loaded uniformly on the GO sheets. The DSC results confirm that the heat release of the nanocomposite with this formulation was 1566.7 J/g, which is approximately four times the heat released by the pure Al–CuO, and the peak temperature dropped by 7.9–26.4 °C. As the GO content increased, the exothermic heat of the composites decreased, indicating that the excess GO played a negative role in the exothermic reaction of the composites. The introduction of GO changed the conventional Al–CuO reaction mechanism, and the reaction produced CuAlO_2_. The burning rate measurement results show that GO inhibits the combustion of Al–CuO. The nanocomposite with 5wt% GO content had a 35.5% lower burning rate than pure Al–CuO. The results show that the self-assembly structure of Al/CuO nanocomposites with GO sheets has the advantages that its exothermic and combustion performances can be modified, which may facilitate the practical application of nanothermites.

## Figures and Tables

**Figure 1 molecules-27-07614-f001:**
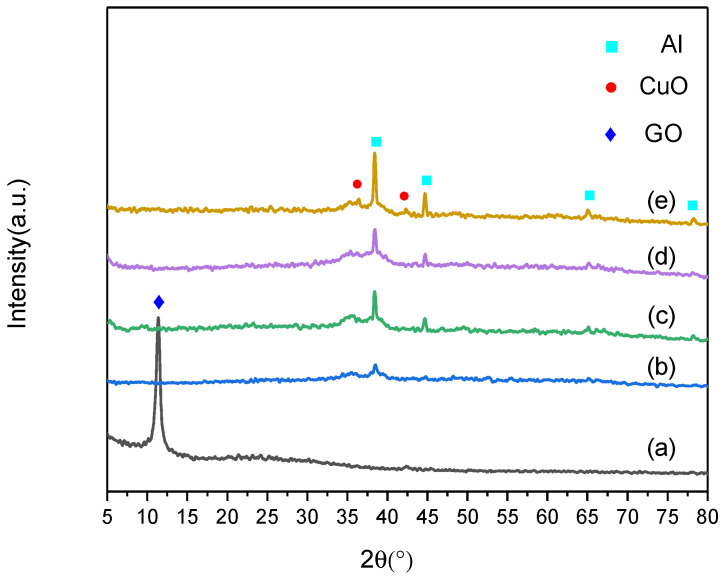
XRD patterns of Al–CuO/GO nanocomposites: (**a**) GO, (**b**) CuO:Al = 2.5:1, (**c**) CuO:Al = 2:1, (**d**) CuO:Al = 1.5:1, and (**e**) CuO:Al = 1:1.

**Figure 2 molecules-27-07614-f002:**
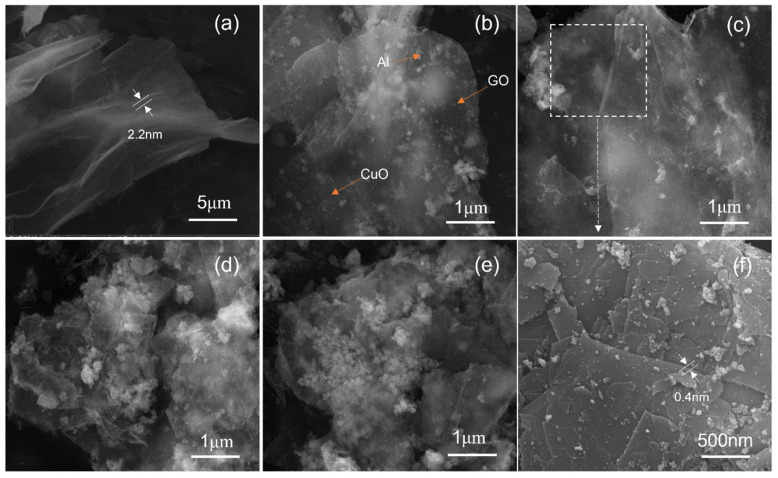
SEM images of Al–CuO/GO nanocomposites: (**a**) GO, (**b**) CuO:Al = 2.5:1, (**c**) CuO:Al = 2:1, (**d**) CuO:Al = 1.5:1, (**e**) CuO:Al = 1:1, and (**f**) a partially enlarged image.

**Figure 3 molecules-27-07614-f003:**
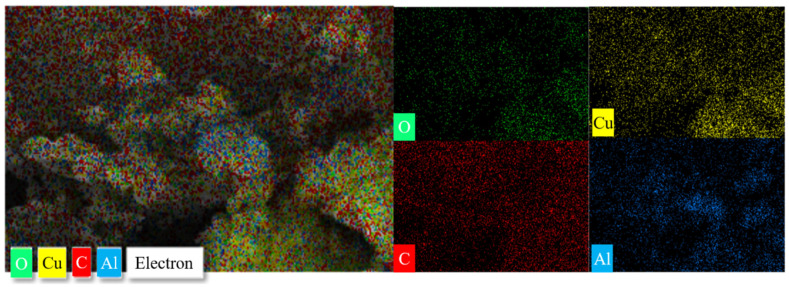
EDS images of Al–CuO/GO with a CuO:Al ratio of 1.5:1.

**Figure 4 molecules-27-07614-f004:**
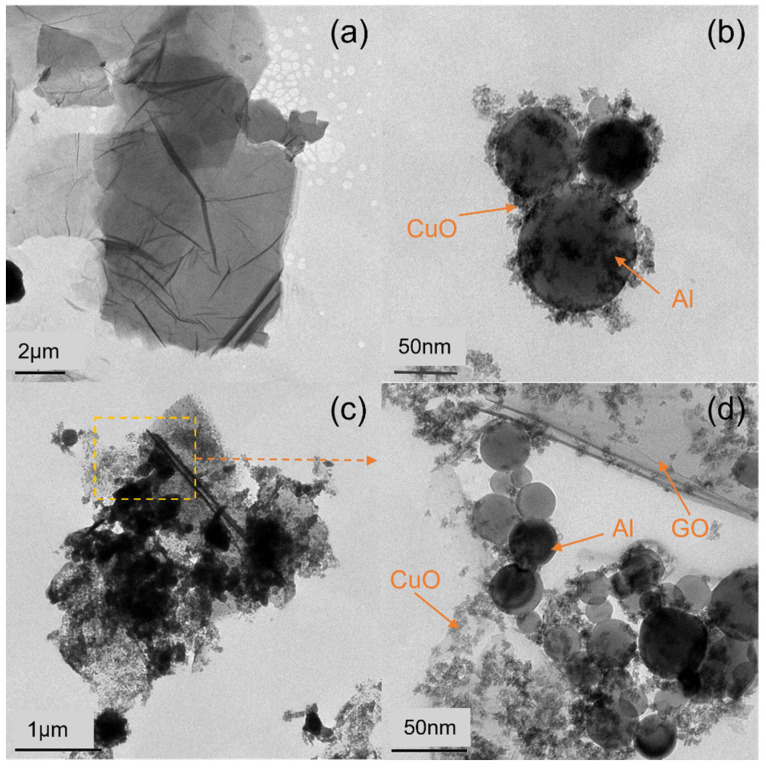
TEM images of (**a**) pure GO, (**b**) pure Al–CuO, (**c**) Al–CuO/GO with CuO:Al = 1.5:1, and (**d**) a partially enlarged image.

**Figure 5 molecules-27-07614-f005:**
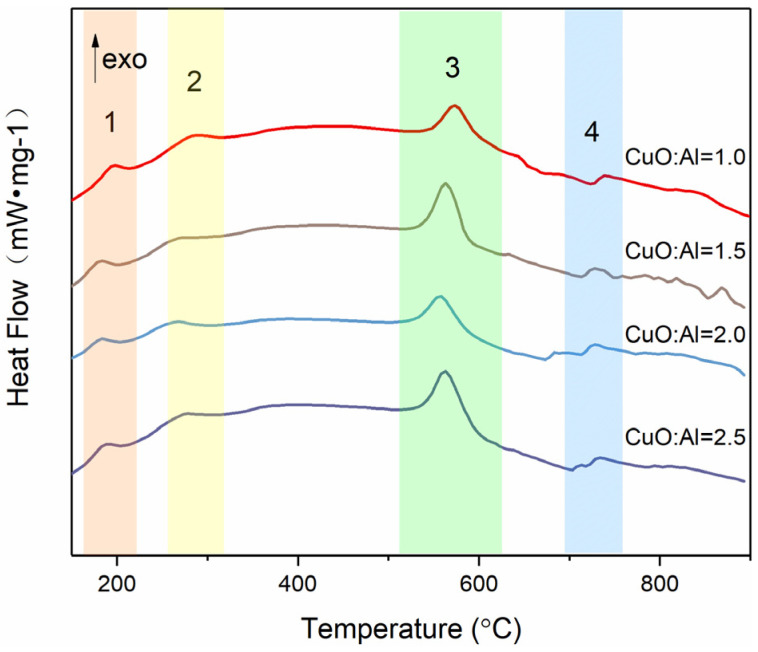
DSC results of Al–CuO/GO MICs with different equivalence ratios of Al and CuO.

**Figure 6 molecules-27-07614-f006:**
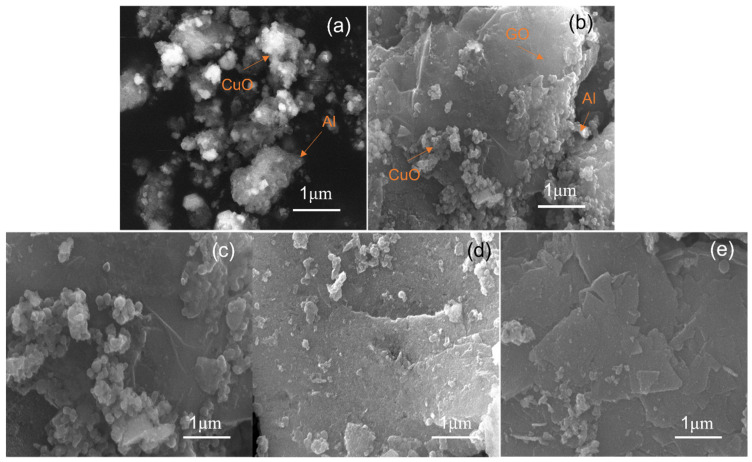
SEM images of Al–CuO/GO with (**a**) 0 wt%, (**b**) 0.5 wt%, (**c**) 1.0 wt%, (**d**) 3.0 wt%, and (**e**) 5.0 wt% of GO.

**Figure 7 molecules-27-07614-f007:**
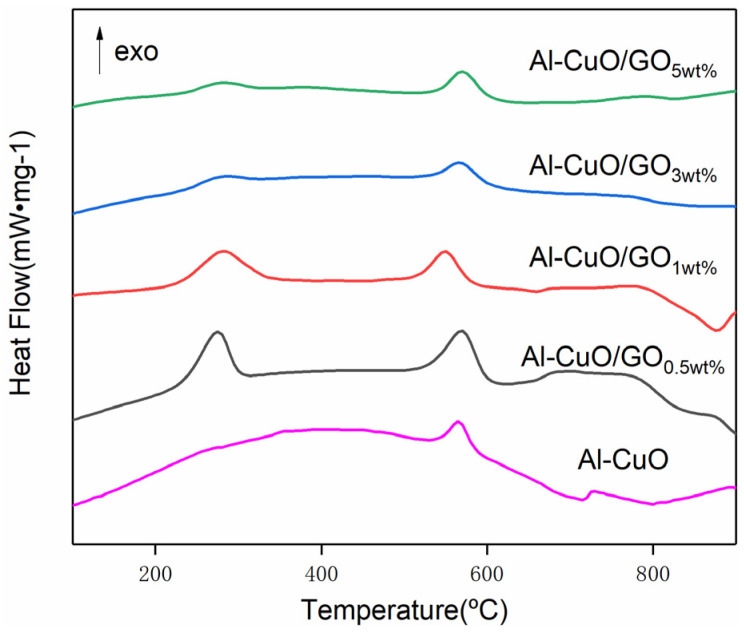
DSC results of composites with 0.5 wt%, 1.0 wt%, 3.0 wt%, and 5.0 wt% of GO at a heating rate of 10 °C/min.

**Figure 8 molecules-27-07614-f008:**
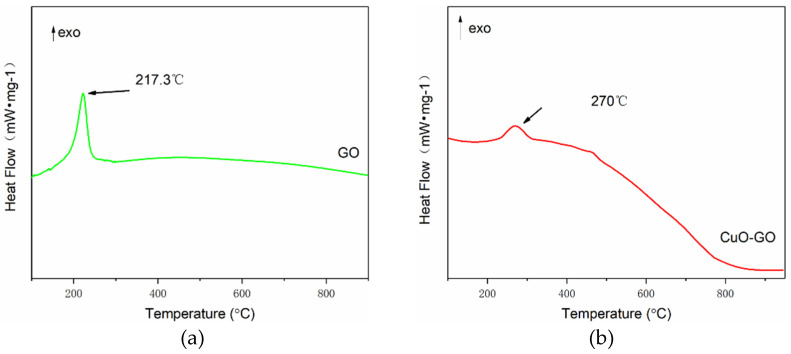
The DSC curves of (**a**) GO and (**b**) CuO–GO.

**Figure 9 molecules-27-07614-f009:**
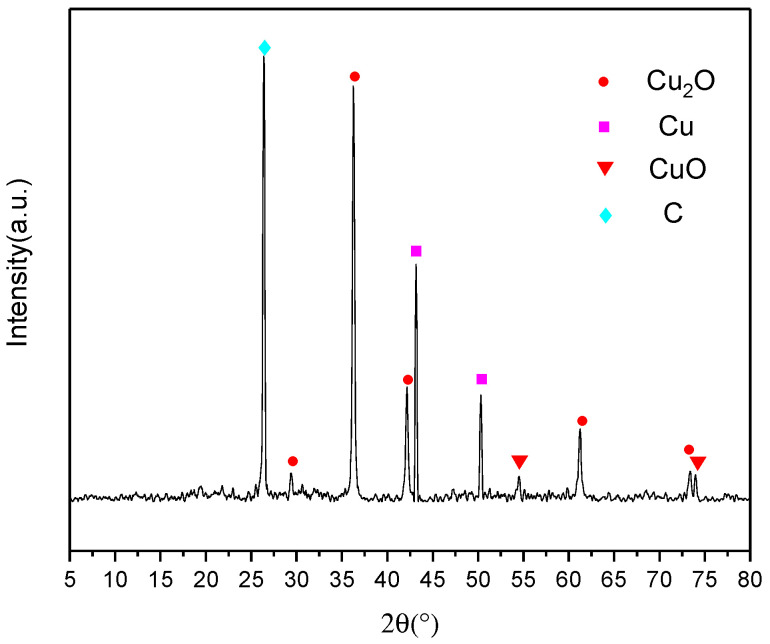
XRD pattern of the reaction products of CuO–GO.

**Figure 10 molecules-27-07614-f010:**
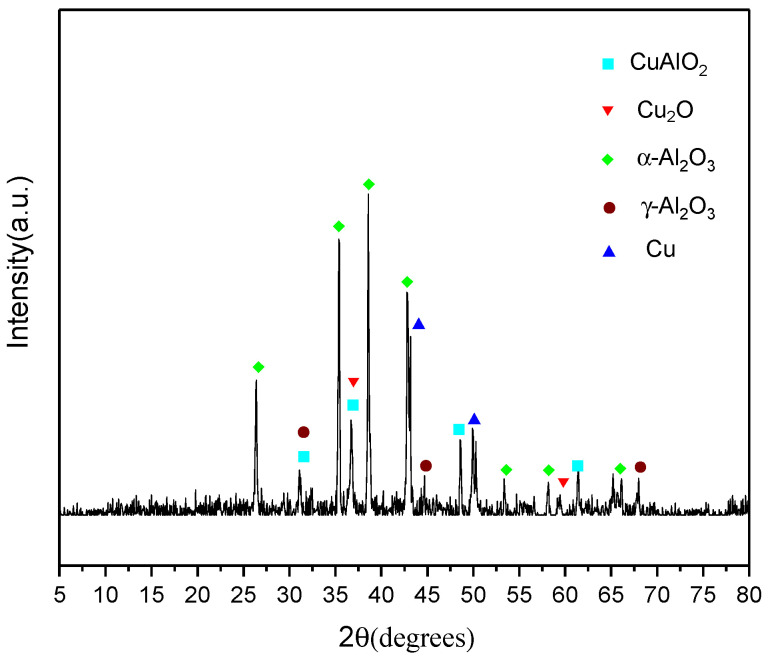
XRD pattern of the reaction products of the nanocomposites (CuO:Al = 1.5:1, GO: 0.5 wt%).

**Figure 11 molecules-27-07614-f011:**
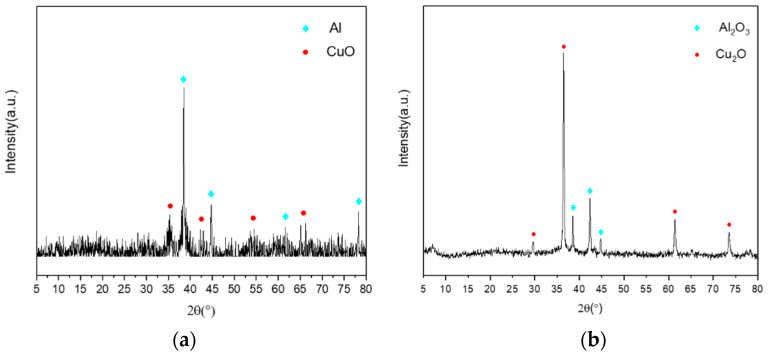
XRD patterns for (**a**) a holding time of 1 h and (**b**) 10 mL of deionized water.

**Figure 12 molecules-27-07614-f012:**
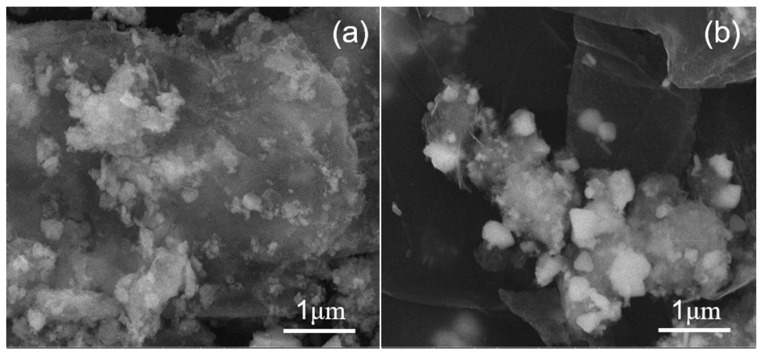
SEM images for (**a**) a holding time of 1 h and (**b**) 10 mL of deionized water.

**Figure 13 molecules-27-07614-f013:**
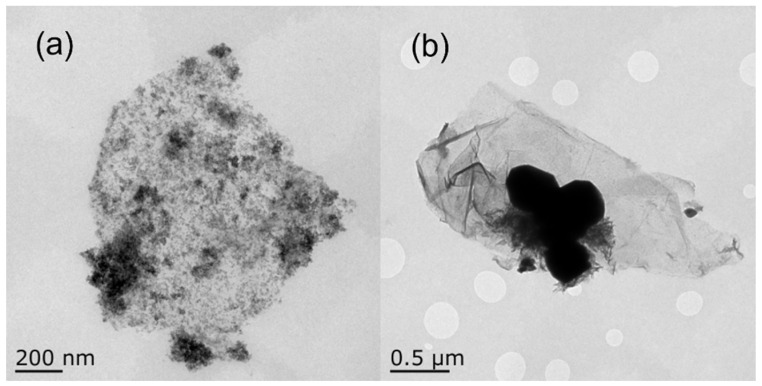
TEM images for (**a**) a holding time of 1 h and (**b**) 10 mL of deionized water.

**Figure 14 molecules-27-07614-f014:**
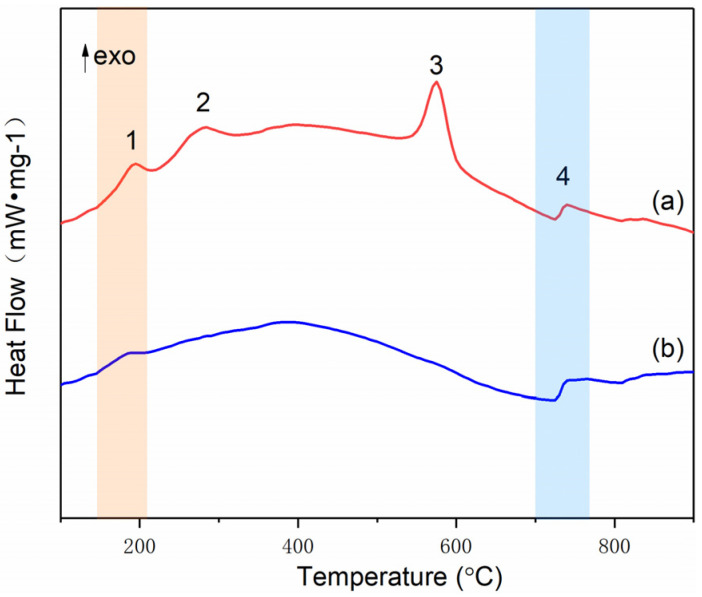
DSC plots for (**a**) a holding time of 1 h and (**b**) 10 mL of deionized water.

**Figure 15 molecules-27-07614-f015:**
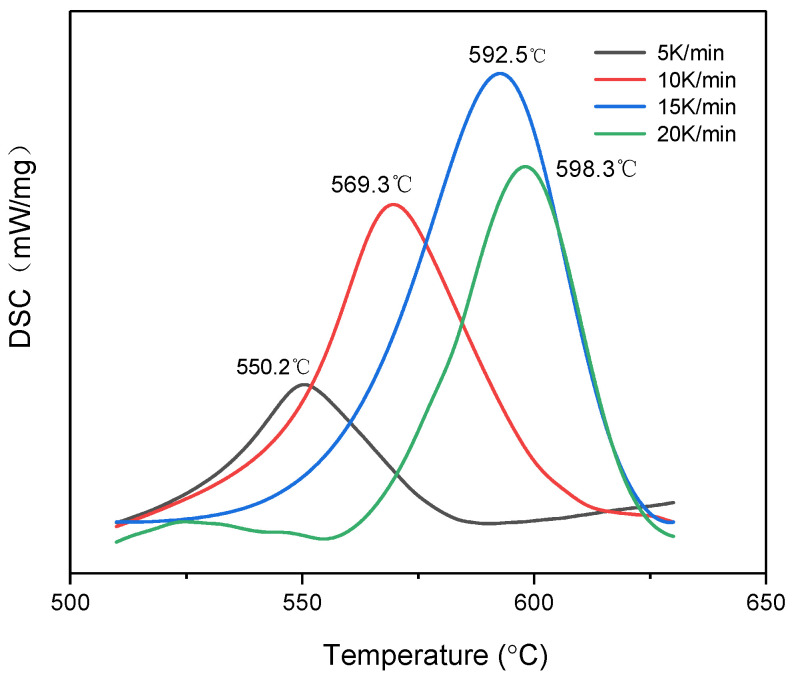
DSC results of Al–CuO/GO at various heating rates.

**Figure 16 molecules-27-07614-f016:**
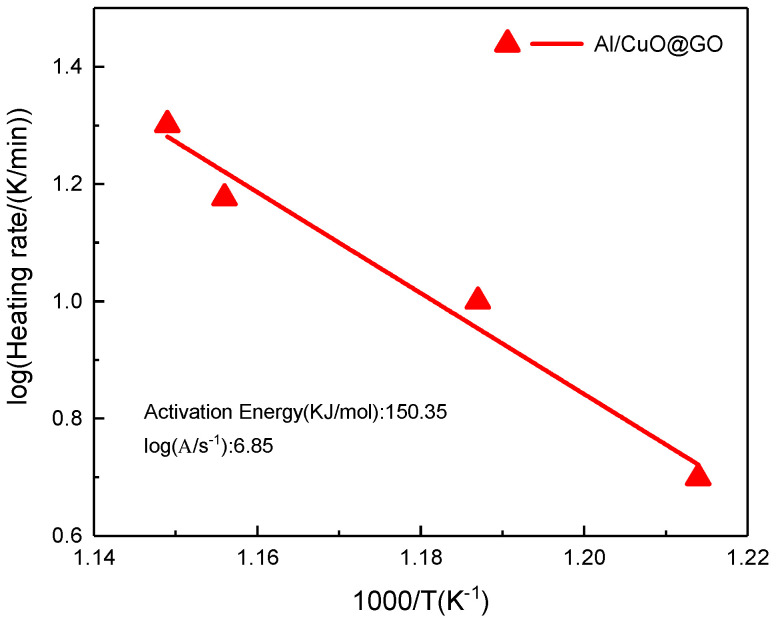
ASTM−698 thermal decomposition fit diagram of Al–CuO/GO at four heating rates.

**Figure 17 molecules-27-07614-f017:**
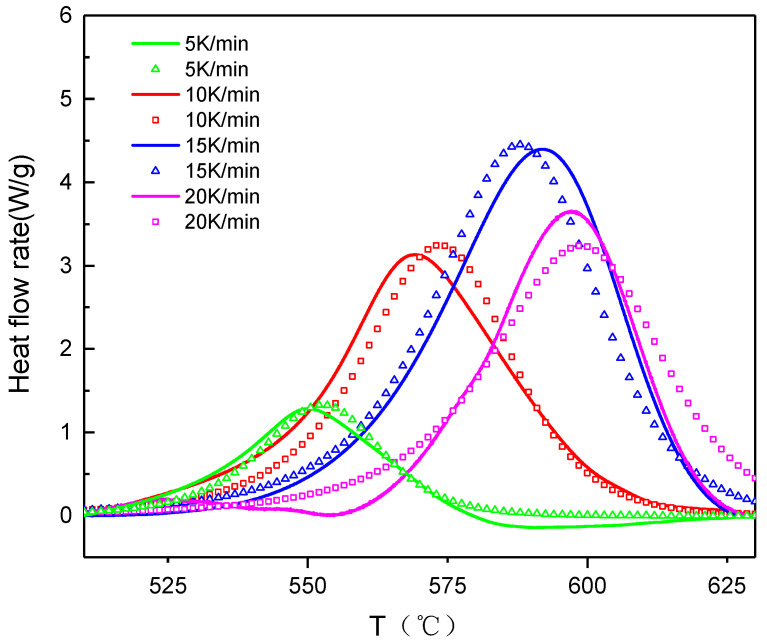
DSC curves of Al–CuO/GO at four heating rates compared between experiments and simulations.

**Figure 18 molecules-27-07614-f018:**
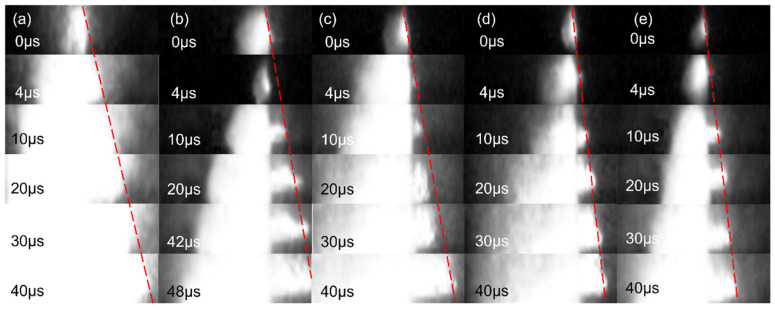
Flame propagation images of (**a**) Al–CuO, (**b**) Al–CuO/GO_0.5wt%_, (**c**) Al–CuO/GO_1.0wt%_, (**d**) Al–CuO/GO_3.0wt%_, and (**e**) Al–CuO/GO_5.0wt%_. The dotted line represents the burning end face.

**Figure 19 molecules-27-07614-f019:**
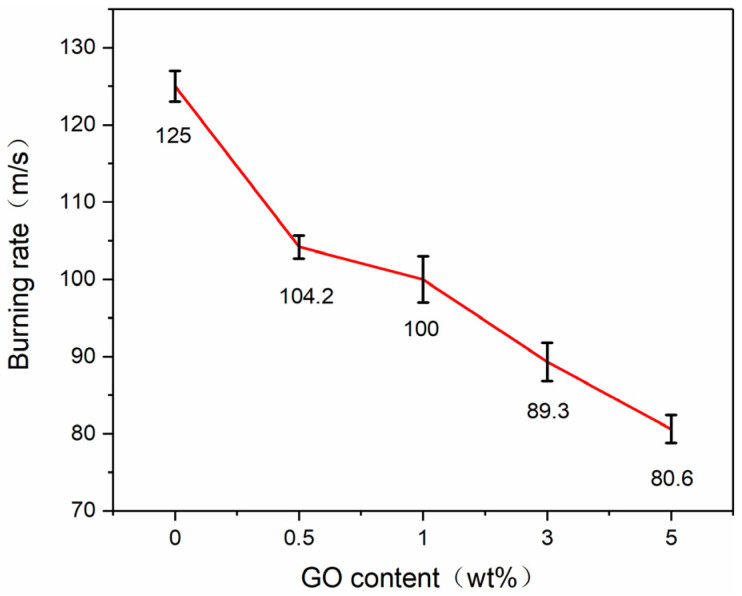
Burning rate of Al–CuO/GO MICs with different GO contents.

**Table 1 molecules-27-07614-t001:** DSC parameters of the Al–CuO/GO MICs at a heating rate of 10 °C/min.

Sample	T_pk1_ (°C)	T_pk2_ (°C)	T_pk3_ (°C)	T_pk4_ (°C)	ΔH (J/g)
CuO:Al = 1	193.1	279.5	568.7	739.5	711.5
CuO:Al = 1.5	190.8	279.4	554.8	737.8	1045
CuO:Al = 2	193.7	275.8	555.2	739.8	886
CuO:Al = 2.5	191.6	278.2	561.5	830.5	811.4

**Table 2 molecules-27-07614-t002:** DSC parameters of Al–CuO/GO with 0 wt%, 0.5 wt%, 1.0 wt%, 3.0 wt%, and 5.0 wt% of GO.

Sample	T_pk1_ (°C)	T_pk2_ (°C)	T_pk3_ (°C)	ΔH (J/g)
Al–CuO	/	576.3	727.2	400.4
Al–CuO/GO_0.5wt%_	278.3	568.4	756.2	1567
Al–CuO/GO_1wt%_	282.2	549.9	782.7	1235
Al–CuO/GO_3wt%_	278.4	567.1	770.9	709.3
Al–CuO/GO_5wt%_	279.9	570.6	789.1	726.6

**Table 3 molecules-27-07614-t003:** DSC parameters of the nanocomposite at different conditions in argon.

Sample	T_pk1_ (°C)	T_pk2_ (°C)	T_pk3_ (°C)	T_pk4_ (°C)	ΔH (J/g)
Holding time is 1 h	191.4	278.1	575.3	898.4	652.3
Water content is 10 mL	181.6	/	/	740.5	115.2

**Table 4 molecules-27-07614-t004:** The characteristic temperatures and exothermic enthalpy of Al–CuO/GO at various heating rates.

β (K·min^−1^)	T_e_ (°C)	T_p_ (°C)	T_f_ (°C)	ΔH (J/g)
5	534.3	550.2	575.1	427.8
10	545.2	569.3	602.1	415.3
15	557.1	592.5	618.4	480.7
20	569.0	598.3	624.1	598.5

**Table 5 molecules-27-07614-t005:** The optimal values of the kinetic factors of Al–CuO/GO obtained using the nonlinear multivariate regression approach.

Kinetic Parameters	Al–CuO/GO
lg A (s^−1^)	4.97
Eα (kJ·mol^−1^)	134.92
lgk_ca_	1.67
Reaction order (n)	1.20
Correlation coefficient	0.974

## Data Availability

Not applicable.

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
