# Peer review of "The Role of Graphene Oxide in the Exothermic Mechanism of Al/CuO Nanocomposites"

_molecules, 2022, doi:10.3390/molecules27217614_

Round 1
Reviewer 1 Report
HRTEM should be carry out to see prefectly Al and and CuO oxide
Reviewer 2 Report
It is very difficult, if not impossible, to review a paper that in most of its parts is incomprehensible. A thorough linguistic review is mandatory. In addition, the authors should strive to adequately describe the results, use clear and systematic notation, and avoid any misleading or confusing notation. For instance, I couldn’t review the analysis of the SEM and TEM micrographs because it was impossible for me to decipher the text. I had the same problem in the discussion of the results at the end of page 9, it was impossible to decipher the text content. Besides the length of the paper is inappropriate, is too long and the analysis that I was able to decipher was quite poor. Therefore, my advice is to reject the article. Authors should produce a fully revised manuscript that is easy to read, with clear notation and clear analysis of the results. Moreover, the analysis of the results need to be significantly improved.
Comments:
1. In lines 80 and 81: the authors state the compositions: “Including the equivalent ratio of CuO/Al (1:1, 1.5:1, 2:1, 2.5:1) and the content of GO (0wt%, 0.5wt%, 81 1wt%, 3wt%, 5wt%).” Is the CuO/Al ratio molar or mass percent?
2. The compositions should all be indicated with the same nomenclature and units. For example, on line 81 the nomenclature is CuO/Al (1:1), in most of the text CuO:Al=1:1, in Figure 5 Al/CuO=1 and in Table 3 CuO:Al=1. And last but not least, what is the meaning of " Al-CuO/GO0.5wt%”? Does the weight percentage only refer to GO or does it also apply to Al and CuO content?
3. In Figures 1 and 2, what is the amount of GO in the samples? Only the relative composition of CuO and Al is indicated.
4. In line 128 the authors state: “No significant Al peaks were detected in the nanocomposites due to the small amount of Al in the system as shown in Figure.1(b). As the Al content grew, the characteristic peaks of Al could be identified.” But, according to the figure caption, it is the amount of CuO that increases, so that the percentage of Al steadily decreases. Therefore, there is an apparent contradiction between the relative increase in the intensity of the aluminum peaks in XRD and the decrease in their relative amount. The same problem appears on page 148 when analyzing the SEM images. Is it possible that the composition is just the opposite of what is indicated in the text? Does this mean that when the authors state CuO:Al=2.5:1 the meaning is that there is 2.5 times more Al than CuO? If so, their entire notation is incorrect.
5. In DSC figures, like figure 5, it is compulsory to indicate with an arrow the direction for exothermic enthalpies.
6. The data in Table 1 do not agree with those in Figure 5. For example, for the composition CuO:Al=1.5 Table 1 gives the highest enthalpy while in Figure 5 the peak is the smallest. The same applies to figure 7 and table 2.
7. Line 212. I do not believe that a DSC can determine an enthalpy with 5 significant figures: 1566.7 J/g.
8. Which is the CuO-Al composition of samples in figure 6.
9. The authors state, “there is an exothermic peak (pink2) at 576.3% and an endothermic peak (pink3) at 727.2℃. The heat 209 release peak corresponds to the thermite reaction, and the endothermic peak is Al melting.” And the exothermic peak for different GO compositions is at 907.2, 880.7, 850.9 and 828.1 ºC. Aluminum melts at 660°C, so it is hard to believe (not to say impossible) that it melts at such high temperatures. Furthermore, measuring a temperature peak with four significant figures is quite unrealistic in a commercial DSC.
10. Line 238. The XRD peak corresponds to graphite no to graphene.
11. There is no evidence that the reaction of equation 1 occurs. There is no evidence that carbon reacts with CuO to form Cu, Cu2O and CuO simultaneously. The presence of carbon facilitates the reduction of copper but the actual reactions resulting in the formation of Cu and Cu2O may be considerably more complex than equation 1.
12. The authors claim that they have applied Kissinger method but they have applied ASTM E-689 method, which is a different method.
13. It is very basic from error analysis that a result cannot be like this one: 150.35±16.89KJ/mol, it should be 170±20 kJ/mol.
Minor comments:
Line 4: The “and” is misplaced.
Line 85. The first time that the acronym Cu(OAc)2•H2O is introduced it is not indicated that corresponds to copper acetate monohydrate.
Line 113. Replace 500000 by 500,000.
Line 223: Replace pink by peak.
Reviewer 3 Report
In this work a series of experimental results are presented on the characterization of graphene oxide/Al/CuO materials and on their thermal behviour, in order to optimize the conditions for the best combustion performance. The work is interesting and the effort of authors to combine several experimental techinques is to be appreciated. However, my opinion is that a thorough revision of the manuscript is required to make it suitable for publication. Besides a number of style and language issues, several scientific questions are raisen from a reading of the manuscript.
In the following, a few specific remarks are reported:
1. The authors’ list ends with an “and”. Please check.
2. Very basic: the carbon-based system used in the work is named “Graphene Oxide” in the title of the manuscript, “graphite oxide” in the introduction: what kind of physical form of carbon we are dealing with in this work ?
3. Many sentences in the text are hard to read, with lacking verbs or other essential parts of the proposition. For the sake of example: page 3, line 98-99 “Transmission electron.....on the sample”; page 5, line 153 “characeterized teh composites with higher....(TEM)”.
4. Since authors’ measurements are based on changing a number of experimental parameters, every time authos present their experimental results, the properties of the studied sample have to be accurately specified. For example, what is the GO amount of samples in Figure 5 ?
5. Many experimental values reported in the work lack of any indication of experimental uncertainty: see for example tables 1, 2. Furthermore, the calibration procedure of temperature and enthalpy for DSC measurements should be reported.
6. At page 8 (line 209), and page 9 (line 233) authors use the expression “pink2”, “pink3”: what is the meaning ?
7. Page 9, line 230: check the word “intional”
8. Page 9, line 233: Fig. 6 should be maybe Fig. 7 ?
9. The control experiments on GO and CuO-Go were carried out only in the limited temperature ranges reported in Fig. 8 ? See, for comparison, the wider T range covered in Figs 5 and 7.
Page 8, line 209: 576.3% should be probably 576.3 °C
Page 14, line 340. The number 150.35 +/- 16.89 seems to be given with too many significant figures.
Page 15, line 352: the reference to Fig. 11 seems wrong.
An accurate revision of the manuscript is strongly recommended, both with regard to the scientific content and in the editorial style. After that, a further peer review is recommended.
Round 2
Reviewer 3 Report
Authors have taken into consideration all the remarks done by this referee. It seems to me that the overall quality of the presentation has significantly improved.
Author Response
Dear Editor and Reviewers:
Thank you for your recognition of our revised manuscript entitled ' The Role of Graphene Oxide in the Exothermic Mechanism of Al/CuO Nanocomposites '. We really appreciate all your comments and suggestions. Your valuable opinions are of great help to the improvement of the quality of our manuscript and work. I must thank you again for your generous help. Everything goes well for you!
Yours sincerely,
Yan Hu